# A multidisciplinary Prematurity Research Cohort Study

**Molly J. Stout[1], Jessica Chubiz[2], Nandini Raghuraman[2], Peinan Zhao[2], Methodius G. Tuuli[3], Lihong V. Wang[4], Alison G. Cahill[5], Phillip S. Cuculich[6], Yong Wang[2], Emily S. Jungheim [7], Erik D. Herzog[8], Justin Fay [9], Alan L. Schwartz[10], George A. Macones[5], Sarah K. England [2]***

1 Department of Obstetrics and Gynecology, University of Michigan, Ann Arbor, Michigan, United States of America, 2 Department of Obstetrics and Gynecology, Washington University in St. Louis, St. Louis, Missouri, United States of America, 3 Department of Obstetrics and Gynecology, Brown University, Providence, Rhode Island, United States of America, 4 Department of Medical Engineering, California Institute of Technology, Pasadena, California, United States of America, 5 Department of Women's Health, University of Texas at Austin, Austin, Texas, United States of America, 6 Department of Internal Medicine, Washington University in St. Louis, St. Louis, Missouri, United States of America, 7 Department of Obstetrics and Gynecology, Northwestern University, Chicago, Illinois, United States of America, 8 Department of Biology, Washington University in St. Louis, St. Louis, Missouri, United States of America, 9 Department of Biology, University of Rochester, Rochester, New York, United States of America, 10 Department of Pediatrics, Washington University in St. Louis, St. Louis, Missouri, United States of America

* englands@wustl.edu

## Abstract

### Background

Worldwide, 10% of babies are born preterm, defined as a live birth before 37 weeks of gestation. Preterm birth is the leading cause of neonatal death, and survivors face lifelong risks of adverse outcomes. New approaches with large sample sizes are needed to identify strategies to predict and prevent preterm birth. The primary aims of the Washington University Prematurity Research Cohort Study were to conduct three prospective projects addressing possible causes of preterm birth and provide data and samples for future research.

### Study design

Pregnant patients were recruited into the cohort between January 2017 and January 2020. Consenting patients were enrolled into the study before 20 weeks' gestation and followed through delivery. Participants completed demographic and lifestyle surveys; provided maternal blood, placenta samples, and cord blood; and participated in up to three projects focused on underlying physiology of preterm birth: cervical imaging (Project 1), circadian rhythms (Project 2), and uterine magnetic resonance imaging and electromyometrial imaging (Project 3).

### Results

A total of 1260 participants were enrolled and delivered during the study period. Of the participants, 706 (56%) were Black/African American, 494 (39%) were nulliparous, and 185 (15%) had a previous preterm birth. Of the 1260 participants, 1220 (97%) delivered a live

**Data Availability Statement:** The data are not publicly available as the minimal data set for this study on pregnant participants contains identifying patient-level data which cannot be suitably de-identified or aggregated. Additionally, a subset of

participants did not consent for future research in the patient consent form approved by the Institutional Review Board at Washington University in St. Louis. Proposals for access to this data should be directed to christinekramer@wustl.edu, Senior Clinical Research Coordinator in the Division of Clinical Research in the Department of Obstetrics and Gynecology. To gain access, data requestors will need to sign a data access agreement.

**Funding:** This work was supported by a research grant from the March of Dimes Foundation (to M.J. S, P.Z, M.G.T., L.V.W., A.G.C, Y.W., E.S.J., E.D.H., J.F, A.L.S., G.A.M. and S.K.E.). The cohort was make possible by support institutional support from St. Louis Children's Hospital, Barnes-Jewish Hospital, and Washington University School of Medicine.

**Competing interests:** The authors have declared that no competing interests exist.

infant. Of the 1220 with a live birth, 163 (14.1%) had preterm birth, of which 74 (6.1%) were spontaneous preterm birth. Of the 1220 participants with a live birth, 841 participated in cervical imaging, 1047 contributed data and/or samples on circadian rhythms, and 39 underwent uterine magnetic resonance imaging. Of the 39, 25 underwent electromyometrial imaging.

## Conclusion

We demonstrate feasibility of recruiting and retaining a diverse cohort in a complex prospective, longitudinal study throughout pregnancy. The extensive clinical, imaging, survey, and biologic data obtained will be used to explore cervical, uterine, and endocrine physiology of preterm birth and can be used to develop novel approaches to predict and prevent preterm birth.

## Introduction

Preterm birth, defined as delivery before 37 weeks' gestation, affects 1 in 10 babies worldwide and is the leading cause of infant mortality [1]. Neonates who survive are at increased risk of lifelong adverse health outcomes [2–4]. This problem is especially notable in St. Louis, Missouri, USA, where 13% of babies are born preterm and racial disparities are pronounced; 11% of white women and over 17% of Black women deliver preterm [5]. Despite decades of research, we have limited understanding of the causes of preterm birth and few strategies to predict or prevent this adverse pregnancy outcome.

In 2013, Lackritz and colleagues argued that preventing preterm birth would require rigorous research to identify the underlying biological and social determinants. Additionally, they argued for development of new tools to monitor pregnancy and identify those at highest risk of preterm birth [6]. To that end, we formed the Washington University in St. Louis Prematurity Research Center in 2014 with funding from the March of Dimes Foundation, St. Louis Children's Hospital, Barnes-Jewish Hospital, and Washington University in St. Louis. The Prematurity Research Center united a multidisciplinary group of investigators including obstetricians, engineers, circadian biologists, and cardiac electrophysiologists to approach preterm birth in novel ways.

The three primary projects of the Prematurity Research Center focused on identifying anatomic, physiologic, and behavioral features that are associated with and can be used to predict preterm birth. The first project, Cervical Imaging, used high-speed functional photoacoustic endoscopy to quantify anatomic changes during cervical remodeling [7, 8]. The second project, Circadian Rhythms, used actigraphy, hormone secretion patterns, and surveys to determine whether disruption in circadian rhythms is a risk factor for preterm birth [5]. The third project, Uterine Electrical Activity, developed a novel imaging system to noninvasively map electrical activity of the uterus during labor contractions [9–12].

To gather data for these projects, we initiated the Prematurity Research Cohort Study to longitudinally follow over 1000 participants from early in pregnancy through delivery. The purpose of this report is to describe the demographics of this cohort and the types of data and biospecimens obtained. In addition, we address the feasibility of conducting a multidisciplinary study in which a diverse cohort of pregnant patients are followed throughout pregnancy.

## Methods

The Prematurity Research Cohort Study was a prospective, longitudinal cohort study performed at Washington University in St. Louis Medical Center between January 2017 and January 2020. A convenience sample size of 1000 participants was chosen as a balance between an aggressive enrollment target given annual delivery volumes, the need to recruit and retain participants in multiple projects, and the varied outcomes assessed in each project. No *a priori* power analysis was conducted. Participants were enrolled in the first or early second trimester and followed through delivery. The study received ethical approval from the Washington University in St. Louis Institutional Review Board. All participants provided written informed consent for collection and use of clinical, biospecimen, imaging, or questionnaire data.

### Inclusion and exclusion criteria

Women were approached for enrollment if they had a singleton pregnancy ≤20 weeks' gestation (determined by best obstetric estimate including last menstrual period or earliest ultrasound dating available) and met the following inclusion criteria: plan to deliver at Barnes-Jewish Hospital, 18 years of age or older, and English speaking. Patients were not eligible if they were incarcerated or conceived via *in vitro* fertilization. If a major fetal anomaly was diagnosed during pregnancy, it was reviewed by a maternal-fetal medicine attending physician. If the anomaly affected gestational age at delivery, the patient was withdrawn from the study.

### Research staff

A large research staff was assembled to support all projects by enrolling participants, performing longitudinal follow-up, scheduling appointments, coordinating research study visits, contacting study participants who missed study visits, collecting data, and managing specimen collection. The research staff comprised one research coordinator, three registered nurses, one sonographer, three research associates, a research lab coordinator, and one research lab assistant. Additionally, eight staff members worked exclusively on the Labor and Delivery floor to support specimen collection and data acquisition at delivery for multiple studies. This team was available 24 hours a day, seven days a week, 365 days a year, and each spent approximately 25% of their effort on the three Prematurity Research Center projects. In addition, two statisticians provided analytic support, and a scientific editor reviewed all scientific presentations and reports.

### Participant recruitment and longitudinal follow-up

Patients were approached for enrollment at their initial prenatal appointment at two obstetric clinics on the Washington University Medical Campus. One clinic primarily serves patients with public health insurance, and the other primarily serves patients with private health insurance. All potential participants were offered enrollment into projects 1 and 2. Participants were seen at study visits longitudinally throughout pregnancy and at delivery. Study visits were scheduled to obtain data and samples in each of the three trimesters (**Fig 1**). All study visits were aligned as much as possible with routine obstetric care to minimize inconvenience to the participants. When additional visits were needed outside of routine medical care visits, appointments were scheduled during routine business hours at the participant's convenience. For follow-up and retention, participants were contacted by phone and/or text messages to remind them of study visits, and transportation was arranged, if needed, to facilitate study participation. Study personnel were available via phone and text during business hours to answer study-related questions.

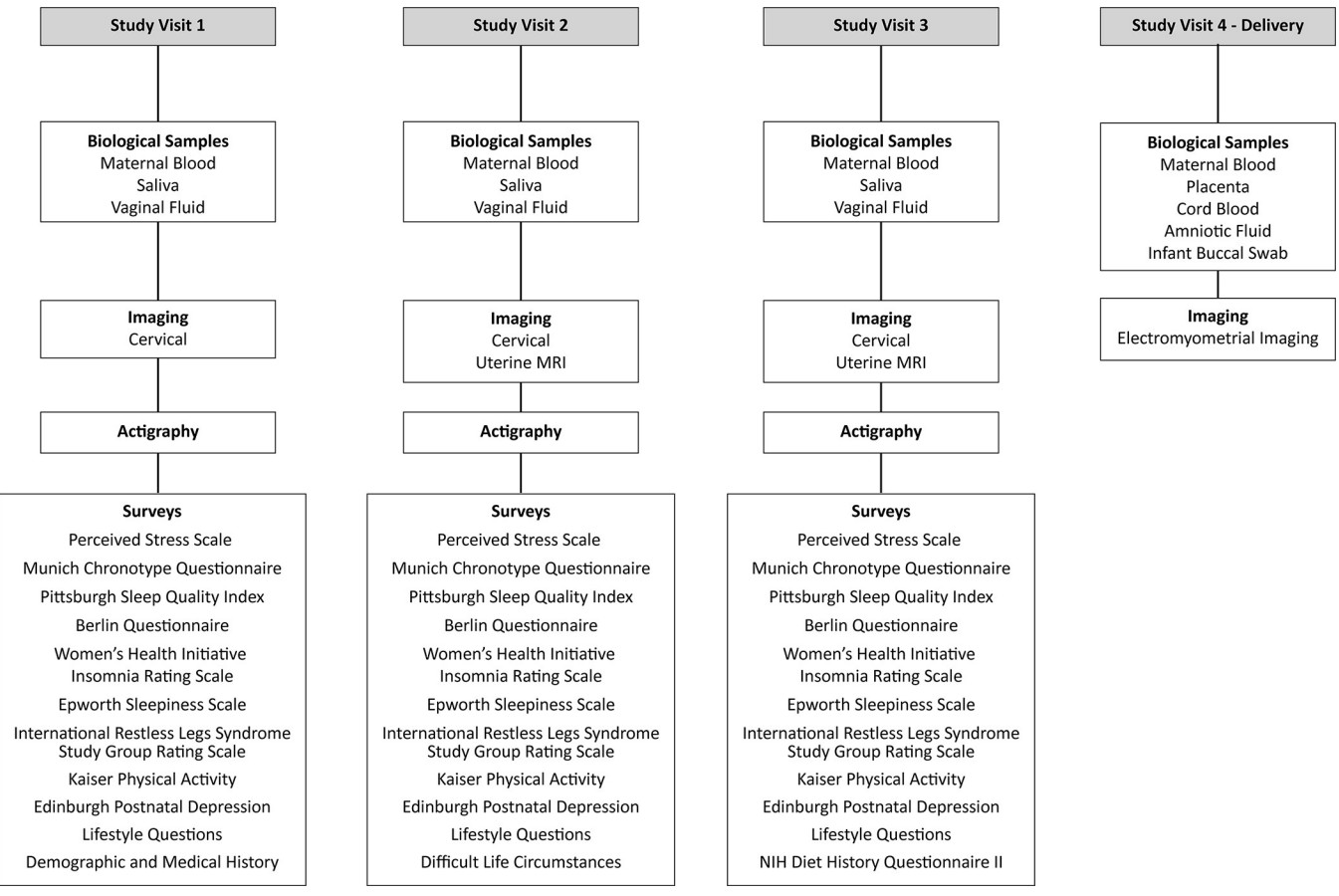

**Fig 1. Timing and sources of data and specimens collected from the cohort.**

## Data and specimen collection

The timing and sources of data and specimens collected are illustrated in **Fig 1**.

**Case report form.** A comprehensive case report form was used to collect data from electronic medical records on the index pregnancy, previous pregnancies, maternal demographics and medical history, labor and delivery, neonatal outcomes until discharge from hospital, and maternal postpartum visits. Obstetric research personnel including research coordinators and research nurses were trained by a Research Nurse coordinator on abstracting clinical data. For quality assurance, the Research Nurse coordinator performed checks on these data. In cases in which the participant delivered at an outside institution, medical record releases were requested to obtain delivery data. The case report form used to collect pregnancy outcome data is shown in **Table 1**.

**Project 1—Cervical imaging.** A research nurse collected swabs from three areas of the vagina (posterior fornix, mid-vagina, and introitus) via speculum exam immediately before performing cervical imaging. Swabs were refrigerated at 4˚C immediately after collection and transferred to -80˚C within 8 hours. After swab collection, novel cervical photoendoscopy devices were used to obtain cervical imaging data transvaginally [7, 8]. Additionally, standard transvaginal ultrasound images of the cervix were obtained and used to measure cervical length at each imaging session. Vaginal swabs were collected and cervical imaging was performed once in each trimester. A subset of participants underwent sampling and imaging up to three additional times, with a minimum of four weeks between sampling/imaging sessions.

**Table 1. Case report form for pregnancy outcomes.**

| |
|---|
| **Date of delivery** |
| **Time of delivery** |
| **Preterm birth (<37 weeks)** |
| **Spontaneous preterm birth** |
| **If spontaneous preterm birth, presentation** |
| Preterm premature rupture of membranes |
| Dilation |
| Preterm premature rupture of membranes + dilation |
| Other (specify) |
| Unknown/not available |
| **Tocolytic medication during pregnancy** |
| None |
| Magnesium |
| Indocin |
| Terbutaline |
| Nifedipine |
| Multiple |
| Unknown/not available |
| **Gestational age at delivery** |
| **If term birth (≥37 weeks), presentation** |
| Premature rupture of membranes |
| Dilation |
| Premature rupture of membranes + dilation |
| Induction |
| Other (specify) |
| Unknown/not available |
| **Was magnesium sulfate administered?** |
| **Clinical chorioamnionitis** |
| **Intrapartum antibiotics** |
| Preterm premature rupture of membranes prophylaxis |
| Group B streptococcus prophylaxis |
| Chorioamnionitis |
| Other |
| Unknown/not available |
| **Type of delivery** |
| Spontaneous vaginal |
| Operative vaginal |
| C-section |
| Unknown/not available |

**Project 2—Circadian rhythms.** All participants completed validated questionnaires about sleep habits and other lifestyle measures (**Fig 1** and **S1 Table**). Participants were given the option of completing surveys via an online link or taking the questionnaires home and returning them to research staff at their next obstetric or research study visit. Research staff called or texted participants to remind them to bring completed surveys to subsequent study visits. Patients with incomplete third trimester surveys were offered the opportunity to complete the surveys on an iPad during admission for delivery.

Participants provided salivary samples every four hours during one 24-hour period each trimester, with collection starting at 18:00 hours. Participants received supplies to collect saliva (Salimetrics, United Kingdom) at home. Research staff called or texted participants to remind them of instructions for collecting saliva and to bring their samples to their next visit. Participants were instructed to place samples in the freezer until bringing them to the research staff. Once received, the samples were timestamped, stored at -80˚C, and processed to measure melatonin and cortisol concentrations by ELISA (Salimetrics Melatonin ELISA kit and Salimetrics Cortisol ELISA kit) in a research laboratory.

Participants wore wrist actigraphy devices (Motionwatch8, CamNTech, United Kingdom) for two-week time periods during their first, second, and third trimesters, as outlined by Martin-Fairey et al. [5]. The actigraphy devices captured minute-level movement and light exposure and remained charged for approximately 90 days, ensuring continuous data collection. Research staff called, texted, or emailed participants to remind them to bring the devices back to the next study visit after the two-week data capture period, or they arranged a courier service for retrieval. If the device was not returned, self-addressed, stamped envelopes were mailed to participants' addresses with a letter requesting return of the device and offering a $20 gift card if they did so.

**Project 3—Uterine electrical activity.** Patients meeting the following inclusion/exclusion criteria were offered enrollment into Project 3: Pre-pregnancy body mass index $< 40 \text{kgm}^2$, willing and able to come to all MRI study appointments, no claustrophobia, no metal implants or non-removable body piercings, and no plans for scheduled cesarean delivery. Sixty-three participants from the total cohort were enrolled to participate in Project 3. Participants included those at low risk for preterm birth (defined as a normal cervical length at anatomy screen and no history of spontaneous preterm birth) and those at high risk for preterm birth (defined as a previous spontaneous preterm birth less than 35 weeks or a cervical length less than 2 cm during the index pregnancy). Those in the low-risk group underwent uterine magnetic resonance imaging (MRI) at 37 weeks' gestation, and those in the high-risk group underwent MRI at 24, 28, and 32 weeks gestation. Once patients presented for induction or in spontaneous labor and were in active labor (defined as greater than 4 cm dilation and regular contractions), body surface potential mapping was performed for approximately one hour. The combined uterine MRI and body surface potential mapping resulted in data used for electromyometrial imaging (EMMI), which has been described elsewhere [9–12]. MRI was performed in a 3T Siemens Prisma/Vida whole-body MRI Scanner with a radial volume interpolated breath-hold examination fast T1-weighted sequence. Patients who also consented to be a part of the cervical imaging project had MRI performed on the same day.

## Biological specimens

Maternal blood serum and plasma samples were collected throughout pregnancy during routine clinical lab visits during business hours (or drawn by research staff if labs were done elsewhere or missed), refrigerated at 4˚C, and centrifuged at 1620 x g for 5 minutes at 4˚C within 12 hours of collection. Aliquots (1 mL) were stored at -80˚C. Cord blood serum and plasma were collected at delivery and processed in the same manner as the maternal blood. In cases in which cord blood was not collected, infant buccal swabs were collected with the mother's consent. At least 30 minutes after the infant was fed, a swab was rubbed firmly against the inside cheek and lower and upper lip for one minute and stored at room temperature. Four sets of placenta specimens (1x3 cm) were collected at delivery from each of four sites: chorionic amnion, basal plate, villous tissue, and subchorion. All placental samples were snap frozen in liquid nitrogen and stored at -80˚C. Amniotic fluid was collected at the time of delivery from

56 patients who underwent unlabored, intact cesarean section. Fluid was centrifuged at 1620 x g for 5 minutes at 4˚C and then stored at -80˚C.

## Participant incentives

All participants received gift cards for completing study visits. Participants received $25 gift cards each trimester for completing the combination of surveys, wrist-worn actigraphy, and 24-hour saliva collection. Participants also received $25 gift cards for each completed transvaginal imaging exam, $50 gift cards for each MRI, and a $50 gift card at delivery if the majority of study procedures were completed. Participants also received a small non-monetary gift (e.g., pen, reusable bag, children's book) at the completion of each study visit and at delivery. For participants without reliable transportation, taxis or Uber Health cars were arranged for transport to and from study appointments at no cost to the participant. For prolonged study visits that spanned a mealtime (typically 3+ hours; combining clinical appointment with transvaginal imaging and MRI), a meal was provided to the participant. Granola bars and other small snacks were available to participants for shorter study visits. Crayons and coloring pages were offered to the children of participants who attended study visits.

# Results

## Participant attrition and demographics

A flow diagram of participant enrollment and outcomes is provided in **Fig 2**. We screened 7478 patients for potential enrollment; 2718 (36.3%) met inclusion criteria (<20 weeks' gestation, age >18 years, English speaking) and were approached. Among those approached, 1523

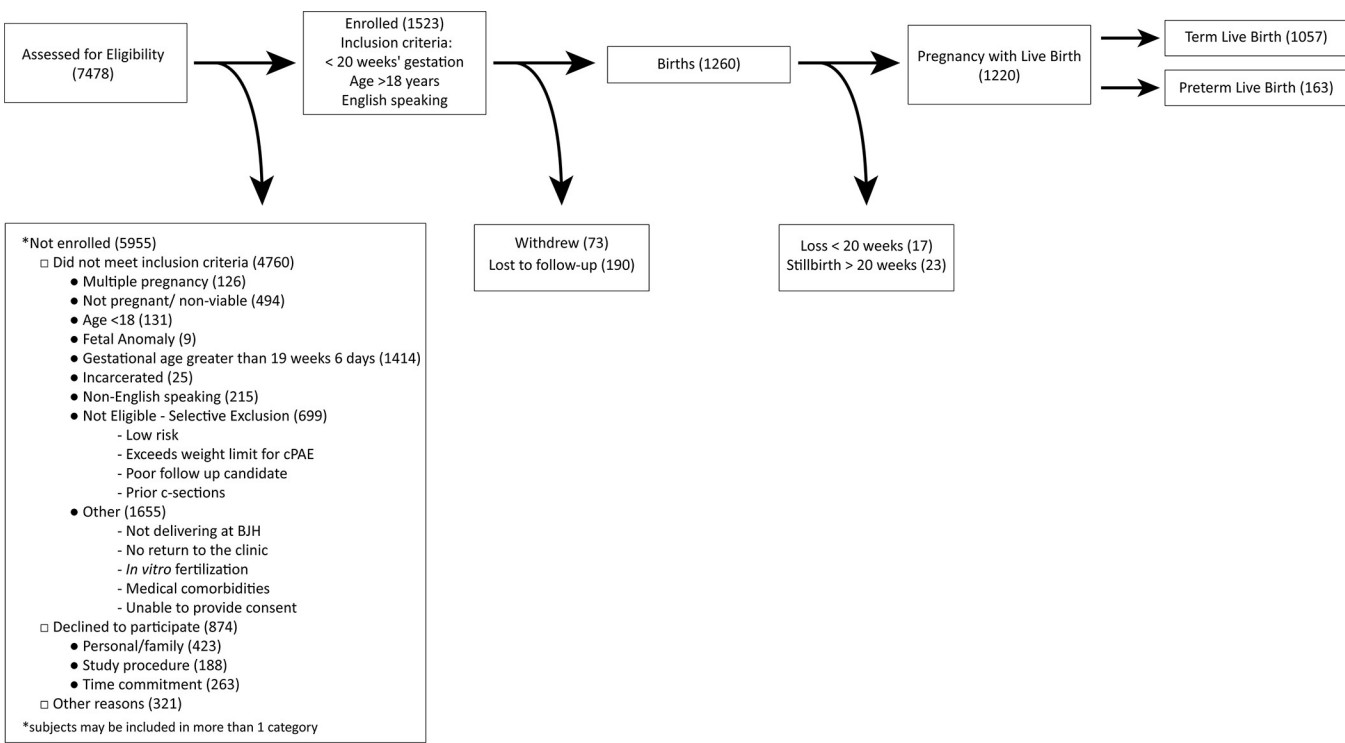

**Fig 2. Flow diagram of participant enrollment and outcomes.** cPAE (cervical photoacoustic endoscopy).

**Table 2. Demographic data of study participants.**

| Total enrolled with outcome data | N = 1260 |
|---|---|
| **Education** | |
| Less than 12<sup>th</sup> grade | 77 (6.1%) |
| High school degree/GED | 583 (46.3%) |
| College degree (4 years) | 179 (14.2%) |
| Graduate degree | 250 (19.8%) |
| Missing/Unknown | 171 (13.6%) |
| **Marital status** | |
| Single | 771 (61.2%) |
| Married | 460 (36.5%) |
| Other | 29 (2.3%) |
| **Employment** | |
| Yes | 892 (70.8%) |
| No | 273 (21.7%) |
| Student | 39 (3.1%) |
| Missing/Unknown | 56 (4.4%) |
| **Annual income (T1)** | |
| Government Assistance | 92 (7.3%) |
| <$25,000 | 465 (36.9%) |
| $25,000-$74,999 | 265 (21.0%) |
| $75,000-$124,999 | 155 (12.3%) |
| ≥$125,000 | 160 (12.7%) |
| Missing/Unknown | 123 (12.7%) |
| **Insurance** | |
| Medicaid | 410 (32.5%) |
| Medicare | 21 (1.7%) |
| Individual/Group Health Insurance | 679 (53.9%) |
| VA/Military | 12 (1.0%) |
| Uninsured | 134 (10.6%) |
| Missing/Unknown | 4 (0.3%) |
| **Race** | |
| Black or African-American | 706 (56.0%) |
| White | 501 (39.8%) |
| Other | 53 (4.2%) |
| **Ethnicity** | |
| Non-Hispanic | 1213 (96.3%) |
| Hispanic | 36 (2.9%) |
| Unknown | 11 (0.9%) |
| **English first language** | **1251 (99.3%)** |

(56.0%) gave consent and enrolled. Of the enrolled participants, 190 (12.5%) were lost to follow-up and 73 (4.8%) withdrew and were not included in the final analyses. We have complete clinical and outcome data on 1260 (82.7%) participants.

Demographic characteristics of the 1260 participants are described in **Table 2**. The majority of participants reported being employed (70.8%), Black/African American (56.0%), and single (61.2%). More than one-third of participants (36.9%) reported an annual income <$25,000, 44.8% had public insurance or were uninsured, and 53.9% had private insurance.

**Table 3. Maternal and pregnancy characteristics.**

| **Gravidity** | |
| --- | --- |
| Gravida, median (IQR) | 2 (1,4) |
| **Parity** | |
| Nulliparous | 494 (39.2%) |
| Multiparous | 766 (60.8%) |
| **History of preterm birth** | |
| Indicated | 10 (0.8%) |
| Spontaneous | 175 (13.9%) |
| 24–33 6/7 weeks | 107 (8.5%) |
| 34–36 6/7 weeks | 68 (5.4%) |
| **History of pregnancy and medical complications** | |
| Asthma | 248 (19.7%) |
| Gestational hypertension/Preeclampsia | 214 (17.0%) |
| Chronic hypertension | 144 (11.4%) |
| Diabetes | 71 (5.6%) |
| Intrauterine growth restriction | 60 (4.8%) |
| Thyroid disease | 57 (4.5%) |
| Anomaly | 47 (3.7%) |
| Heart disease | 37 (2.9%) |
| Renal disease | 29 (2.3%) |
| Polyhydramnios | 22 (1.7%) |
| Oligohydramnios | 10 (0.8%) |
| Lupus | 6 (0.5%) |

Except where noted, data are presented as n (%); IQR, interquartile range

## Pregnancy characteristics and outcomes

Table 3 demonstrates maternal and pregnancy characteristics in our cohort. The majority (n = 766, 60.8%) were multiparous, and 185 (14.7%) had a history of preterm birth. The most common medical complications among these participants was asthma and chronic hypertension. The most common previous pregnancy complications were gestational hypertension/preeclampsia and intrauterine growth restriction (Table 3).

In the cohort of 1260 participants, there were 1220 (96.8%) live births, 23 stillbirths $\geq 20$ weeks, and 17 pregnancy losses <20 weeks (Table 4). Among the 1220 participants with a live birth, 529 (43.3%) participants underwent induction of labor, and 368 (30.2%) were delivered via cesarean section. Among the 1220 participants with a live birth, 1057 (86.6%) delivered at term and 163 (13.4%) delivered preterm (< 37 weeks). Of the live-born neonates, 145 (11.9%) required neonatal intensive care unit admission, 145 (11.9%) had low birthweight (<2500 g), and 153 (13.1%) were small for gestational age, defined as birthweight <10[th] percentile for gestational age.

## Data and specimen collection associated with study procedures

Collectively, study participants attended 6135 study visits, which included 1892 cervical imaging exams, 39 uterine MRIs, 2239 actigraphy recordings, 28,240 lifestyle surveys, and over 12,000 biological specimens (see Table 5). Fewer participants provided saliva samples in the first trimester than in the second and third trimesters. Conversely, more participants completed lifestyle surveys and provided actigraphy data in the first trimester than in the second

**Table 4. Pregnancy and neonatal outcomes.**

| | |
|---|---|
| **Birth outcome** | |
| Live birth | 1220 (96.8%) |
| Loss, <20 weeks | 17 (1.3%) |
| Loss, 20+ weeks | 23 (1.8%) |
| **Gestational age at birth** | |
| <24 weeks | 3 (0.2%) |
| 24 weeks-31 weeks 6 days | 32 (2.6%) |
| 32 weeks-36 weeks 6 days | 128 (10.5%) |
| 37+ weeks | 1057 (86.6%) |
| **Preterm Birth** | |
| Induced | 89 (7.3%) |
| Spontaneous | 74 (6.8%) |
| x00A0; <24 weeks | 2 (.002%) |
| x00A0; 24–33 weeks 6 days | 22 (1.8%) |
| x00A0; 34–36 weeks 6 days | 50 (4.1%) |
| **Delivery method** | |
| Vaginal | 806 (66.0%) |
| Operative vaginal | 46 (3.8%) |
| Cesarean section | 368 (30.2%) |
| **Reason for induction (N = 529)**[*] | |
| Oligohydramnios | 6 (1.1%) |
| PROM | 8 (1.5%) |
| PPROM | 7 (1.3%) |
| Preeclampsia/eclampsia | 59 (11.2%) |
| Comorbidity at 39 weeks | 35 (6.6%) |
| Elective | 236 (44.6%) |
| Non-reassuring antenatal testing | 40 (7.6%) |
| Gestational Diabetes | 16 (3.0%) |
| Postdates | 64 (12.1%) |
| Intrauterine growth restriction | 46 (8.7%) |
| Macrosomia | 2 (0.4%) |
| polyhydramnios | 4 (0.8%) |
| Other | 76 (14.4%) |
| **Neonatal Sex** | |
| Female | 581 (47.6%) |
| Male | 639 (52.4%) |
| **Apgar score at 1 minute** | |
| 0–3 | 59 (4.8%) |
| 4–6 | 83 (6.8%) |
| 7–10 | 1053 (86.3%) |
| **Apgar score at 5 minutes** | |
| 0–3 | 8 (0.7%) |
| 4–6 | 40 (3.3%) |
| 7–10 | 1163 (95.3%) |
| **NICU information** | |
| NICU admission | 145 (11.9%) |
| Length of NICU stay in days | 8 (4, 21) |
| **Neonatal health outcomes** | |

(*Continued*)

**Table 4.** (Continued)

| | |
|---|---|
| Low birth weight (<2500 grams) | 145 (11.9%) |
| Small, for gestational age (<10th percentile) | 153 (13.1%) |
| Arterial umbilical cord pH | 7.26 (7.22, 7.31) |
| Newborn death within 28 days | 8 (0.7%) |

*The percentage of reason for induction is calculated according to the total number of inductions. Some participants had more than one reason for induction.

Data represent n (%) or median (interquartile range).

and third trimesters. More patients underwent transvaginal imaging, which required additional scheduling after consent, in the first trimester than in the second and third trimesters.

A total of 63 participants provided consent to undergo uterine MRI and electrical mapping of the uterus (electromyometrial imaging [EMMI]) at labor (**Table 6**). A total of 24 (17 low-risk, 7 high-risk) participants withdrew before the MRI for reasons such as delivery before MRI (8), lost contact (6), and patient/family request to withdraw (6) (**Table 6**). A total of 25 participants in the low-risk group and 14 in the high-risk group underwent MRI at least once during pregnancy (**Table 7**). In addition, 20 participants in the low-risk group and 5 in the

**Table 5. Sample and survey numbers from participants with live births.**

| Samples and Data Collected | First Study Visit | Second Study Visit | Third Study Visit | Delivery |
|---|---|---|---|---|
| **Biologic Samples** | | | | |
| Maternal blood | 795 | 855 | 864 | 1023 |
| Saliva | 453 | 822 | 778 | - |
| Vaginal swabs | 212 | 791 | 588 | - |
| Maternal buffy coat | 790 | 855 | 863 | 430 |
| Placenta | - | - | - | 1047 |
| Cord blood | - | - | - | 870 |
| Cord blood buffy | - | - | - | 864 |
| Amniotic fluid | - | - | - | 56 |
| Infant buccal swab | - | - | - | 211 |
| **Cervical Imaging** | 673 | 629 | 590 | n/a |
| **Actigraphy** | 737 | 806 | 696 | n/a |
| **Surveys** | | | | |
| Perceived Stress Scale | 1136 | 786 | 825 | n/a |
| Munich Chronotype Questionnaire | 1047 | 784 | 823 | n/a |
| Pittsburgh Sleep Quality Index | 1039 | 783 | 822 | n/a |
| Berlin Questionnaire | 984 | 781 | 823 | n/a |
| Women's Health Initiative Insomnia Rating Scale | 983 | 782 | 821 | n/a |
| Epworth Sleepiness Scale | 1035 | 783 | 821 | n/a |
| Intern. Restless Legs | 1032 | 783 | 821 | n/a |
| Kaiser Physical Activity | 1030 | 781 | 821 | n/a |
| Edinburgh Postnatal Depression | 1212 | 874 | 866 | n/a |
| Demographic/Med. Hist. | 1111 | 801 | 845 | n/a |
| Difficult Life Circumstances | n/a | n/a | n/a | 654 |
| NIH Diet Questionnaire | n/a | n/a | n/a | 751 |

A total of 1220 participants had live births.

**Table 6. Project 3 participants.**

| Cohort | Consented | Withdrew | Reasons for Withdrawal | Underwent MRI | Missed EMMI | Reasons for Missed EMMI | Underwent MRI and EMMI |
|--------|-----------|----------|------------------------|---------------|-------------|-------------------------|------------------------|
| **Low Risk** | 42 | 17 | Delivered before MRI (7)<br>Per request (5)<br>Lost contact (4)<br>Medical issue (1) | 25 | 5 | Missed by L&D staff (3)<br>Delivered precipitously (2) | 20 (0 preterm, 20 term) |
| **High Risk** | 21 | 7 | Lost contact (2)<br>Delivered before MRI (1)<br>Lethal anomaly (1)<br>Spontaneous abortion (1)<br>Per request (1)<br>Social issues (1) | 14 | 9 | Delivered at offsite hospital (3)<br>COVID (2)<br>Emergency cesarean (2)<br>Intrauterine Fetal Demise (1)<br>Delivered precipitously (1) | 5 (2 preterm, 3 term) |

EMMI, electromyometrial imaging; L&D, Labor and Delivery; MRI, magnetic resonance imaging

high-risk group underwent EMMI at labor (**Tables 6 and 7**). Reasons for missed EMMI included: not notified by Labor and Delivery staff (3), delivered at offsite hospital (3), COVID-19 research shut-down (2), and emergent cesarean section (2). Among those who underwent both MRI and EMMI, none of the low-risk participants and two of the high-risk participants delivered preterm (**Table 6**).

## Discussion

The Prematurity Research Cohort Study was a multi-faceted study aimed at identifying mechanisms underlying preterm birth. This report demonstrates the feasibility of conducting a longitudinal study in pregnant participants and maintaining high consent and retention rates. Moreover, we describe the rich data and specimen source now available for longitudinal studies of pregnancy.

The primary intent of establishing this cohort was to identify causes of, and develop novel diagnostics to predict, preterm birth. Analyses of data from surveys, swabs, specimens, and imaging are ongoing for the three projects. Specimens are also banked for future research to identify both risk factors and potential biomarkers. The data and specimens we collected will be useful for addressing maternal and neonatal health disparities. This is because over 50% of the participants were Black, and all lived in the St. Louis, Missouri, area, where Black women have a 50% higher risk of preterm birth than white women [5].

### Feasibility

Within three years, we enrolled 1260 participants, and we performed 6135 study visits spanning all trimesters. Notably, 977 women attended three or more visits over the course of

**Table 7. Project 3 procedures performed.**

| Group and procedure | Timing | | | | |
|---------------------|--------|--------|--------|--------|-------|
| | 24 weeks | 28 weeks | 32 weeks | 37 weeks | Labor |
| Low Risk—Uterine MRI | - | - | - | 25 | |
| High Risk—Uterine MRI | 14 | 9 | 9 | - | |
| Low Risk—EMMI | - | - | - | - | 20 |
| High Risk—EMMI | - | - | - | - | 5 |

EMMI, electromyometrial imaging; MRI, magnetic resonance imaging

pregnancy. We collected over 12,000 biological specimens with linked clinical and imaging data. Among the 1260 enrolled participants, 859 (68.1%) participated in Project 1, Project 2, or both, the two projects open to all participants.

Study participants were more likely to comply with study procedures that could be timed with clinical appointments or labs (e.g., blood draws, surveys done in waiting room or exam room). Fewer data and specimens were collected in the third trimester than in the first and second trimesters. In part, this was because some participants delivered before their third trimester study visit.

### Scientific implications

The Prematurity Research Cohort Study has generated a rich set of data and specimens that can be used to test hypotheses pertaining to mechanisms of preterm birth and preventive targets. This dataset and specimen bank will also allow investigators to explore new questions regarding preterm birth, use imaging and biomarkers to assess preterm birth risk in the first trimester, and identify modifiable lifestyle factors that increase risk of preterm birth [6]. Our cohort was predominantly those with the highest risk of preterm birth: women of color and women with socioeconomic stressors. Our cohort had a higher percentage of African Americans (56%) than the percentage in the US population (13.4%) and had high rates of several chronic conditions, which likely reflects the fact that the study was conducted at a tertiary care hospital.

Our experience reveals that pregnant patients are willing to participate in studies that require multiple research visits, undergo serial transvaginal ultrasounds or MRI, answer extensive surveys, wear actigraphy monitors, and collect timed saliva samples and other biological specimens beyond those required for routine clinical care. Thus, other researchers should recognize that pregnant patients both can and should be included in studies for obstetric and non-obstetric outcomes.

### Cost and resource utilization

Conducting this study required extensive investments of time and financial resources. Full or partial salaries were required for the 80 staff members, 31 trainees, and 23 faculty members who participated in various aspects of the study. In addition, the study required abundant supplies (e.g., blood and placenta sample collection tubes), dedicated freezer space, gift cards, and transportation arrangements. Such costs and resources are important to consider in planning any longitudinal study in pregnancy. To reduce costs, future studies could maximize use of web-based and social media tools to optimize recruitment and retention [13, 14]. Although we did not formally assess patient-reported acceptability of participating in the cohort study, future work should investigate barriers and facilitators to patient adherence to multiple research visits during pregnancy.

In summary, we report successful enrollment and follow-up of a large longitudinal cohort of pregnant patients. Additionally, we describe the substantial investments made by participants and research personnel to collect data and specimens. We are optimistic that their contributions will lead to new discoveries to improve the health of pregnant patients and their babies.

### Supporting information

**S1 Table. Sleep and lifestyle data surveys collected from participants.**
(DOCX)

**S1 File.**
(DOC)

## Acknowledgments

We thank the Prematurity Research Cohort Study participants for their invaluable contributions to preterm birth research. We thank the research staff for their tireless efforts enrolling and following participants and collecting and managing data and specimens. We thank Deborah Frank, PhD, Stephanie Pizzella, Christine Kramer, and Jillian Ashley-Martin, PhD, for editorial comments and Anthony Bartley for graphical assistance.

## Author Contributions

**Conceptualization:** Peinan Zhao, Methodius G. Tuuli, Lihong V. Wang, Alison G. Cahill, Emily S. Jungheim, Erik D. Herzog, Justin Fay, Alan L. Schwartz, George A. Macones, Sarah K. England.

**Data curation:** Peinan Zhao.

**Formal analysis:** Molly J. Stout, Nandini Raghuraman, Peinan Zhao, Sarah K. England.

**Funding acquisition:** Methodius G. Tuuli, Lihong V. Wang, Alison G. Cahill, Phillip S. Cuculich, Yong Wang, Emily S. Jungheim, Erik D. Herzog, Justin Fay, Alan L. Schwartz, George A. Macones, Sarah K. England.

**Investigation:** Molly J. Stout, Methodius G. Tuuli, Phillip S. Cuculich, Emily S. Jungheim, Erik D. Herzog, Justin Fay, Sarah K. England.

**Methodology:** Molly J. Stout, Jessica Chubiz, Methodius G. Tuuli, Lihong V. Wang, Alison G. Cahill, Yong Wang, Emily S. Jungheim, Erik D. Herzog, Justin Fay, George A. Macones, Sarah K. England.

**Project administration:** Molly J. Stout, Jessica Chubiz, Methodius G. Tuuli, Lihong V. Wang, Alison G. Cahill, Yong Wang, Erik D. Herzog, Justin Fay, Sarah K. England.

**Supervision:** George A. Macones.

**Writing – original draft:** Molly J. Stout, Jessica Chubiz, Nandini Raghuraman, Sarah K. England.

**Writing – review & editing:** Molly J. Stout, Jessica Chubiz, Nandini Raghuraman, Peinan Zhao, Lihong V. Wang, Alison G. Cahill, Emily S. Jungheim, Erik D. Herzog, Justin Fay, Alan L. Schwartz, George A. Macones, Sarah K. England.

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
