## [Decision Letter · Decision Letter 0]

28 Feb 2022

PONE-D-21-27892A Multidisciplinary Prematurity Research Cohort Study                                                                                    PLOS ONE

Dear Dr. Sarah England,

Thank you for submitting your manuscript to PLOS ONE. After careful consideration, we feel that it has merit but does not fully meet PLOS ONE’s publication criteria as it currently stands. Therefore, we invite you to submit a revised version of the manuscript that addresses the points raised during the review process.

We look forward to receiving your revised manuscript.

Kind regards,

Alireza Abdollah Shamshirsaz

Academic Editor

PLOS ONE

Journal Requirements:

Additional Editor Comments (if provided):

Thank you, Dr. Sarah England, for submitting your work to PLOS ONE.

Reviewers' comments:

Reviewer's Responses to Questions

**Comments to the Author**

1. Is the manuscript technically sound, and do the data support the conclusions?

Reviewer #1: Yes

Reviewer #2: Yes

2. Has the statistical analysis been performed appropriately and rigorously? 

Reviewer #1: Yes

Reviewer #2: Yes

3. Have the authors made all data underlying the findings in their manuscript fully available?

Reviewer #1: Yes

Reviewer #2: Yes

4. Is the manuscript presented in an intelligible fashion and written in standard English?

Reviewer #1: Yes

Reviewer #2: Yes

5. Review Comments to the Author

Reviewer #1: Congratulations on an enormous logistical undertaking.

How was the cohort size determined?

Why were patients conceived via IVF excluded?

The authors state that all patients were offered enrollment into projects 1 and 2. How were patients selected for project 3?

Reviewer #2: This study presents the methodology in establishing a large cohort to investigate preterm birth.

Overall this is a well written and presents a well-designed and elaborated methodology in recruiting patients to the cohort.

Since this manuscript only presents the methodology and does not reveal any of the actual results of that were investigated I question its interest to the readers.

Numerous studies have been previously published on preterm birth some were prospective randomized and some had large cohorts. I believe the ingenuity of this study lies on the three presented investigated topics and less on the recruitment process.

My main concern is whether publishing a prelim manuscript as presented here is even relevant and wonder if all this should just be embedded in the manuscripts publishing the results.

6. PLOS authors have the option to publish the peer review history of their article (what does this mean?). If published, this will include your full peer review and any attached files.

Reviewer #1: No

Reviewer #2: No

---

## [Author Response · Author response to Decision Letter 0]

7 Apr 2022

We thank the reviewers for the thorough and positive review of the manuscript. We have responded to all the comments below. 

How was the cohort size determined? A convenience sample size of 1000 participants was chosen as a balance between an aggressive enrollment target given annual delivery volumes, the need to recruit and retain participants in multiple projects, and the varied outcomes assessed in each project. No a priori power analysis was conducted. We added this information to the Methods section (Lines 93–96). 

Why were patients conceived via IVF excluded? IVF patients were excluded from this cohort because of the a priori increased risk of preterm birth and pregnancy complications. Further, it was unknown at the time whether the IVF process modified the endocrine physiology of pregnancy, including unknown impacts of fresh and frozen cycles on hormonal states.

The authors state that all patients were offered enrollment into projects 1 and 2. How were patients selected for project 3? Patients meeting the following inclusion/exclusion criteria were offered enrollment into Project 3: Pre-pregnancy body mass index < 40kgm2, willing and able to come to all MRI study appointments, no claustrophobia, no metal implants or non-removable body piercings, and no plans for scheduled cesarean delivery. We added this information to the manuscript (Lines 182-185). 

Reviewer #2: This study presents the methodology in establishing a large cohort to investigate preterm birth. Overall this is a well written and presents a well-designed and elaborated methodology in recruiting patients to the cohort. Since this manuscript only presents the methodology and does not reveal any of the actual results of that were investigated. I question its interest to the readers. Numerous studies have been previously published on preterm birth, some were prospective randomized and some had large cohorts. I believe the ingenuity of this study lies on the three presented investigated topics and less on the recruitment process. My main concern is whether publishing a prelim manuscript as presented here is even relevant and wonder if all this should just be embedded in the manuscripts publishing the results.

We argue that the longitudinal approach and recruitment success in a diverse population warrants a separate manuscript independent of the results. Furthermore, the three projects for which this cohort was formed differ substantially from one another. Thus, our plan is to describe the cohort and recruitment process in this index manuscript and then refer back to this manuscript in all future publications that contain results from the projects and future efforts. We argue this approach is more reader friendly than embedding all the information into future individual manuscripts. This approach has been modeled by several other large-scale and longitudinal cohort studies in pregnancy and women’s health. Examples include PMIDs 34610322, 21819423, and 27459450.

---

## [Decision Letter · Decision Letter 1]

14 Jul 2022

A Multidisciplinary Prematurity Research Cohort Study

PONE-D-21-27892R1

Dear Dr. England,

We’re pleased to inform you that your manuscript has been judged scientifically suitable for publication and will be formally accepted for publication once it meets all outstanding technical requirements.

Kind regards,

Prem Singh Shekhawat, MD

Academic Editor

PLOS ONE

Additional Editor Comments (optional):

Thank you for completing minor revisions to your original submission.

Reviewers' comments:

Reviewer's Responses to Questions

**Comments to the Author**

1. If the authors have adequately addressed your comments raised in a previous round of review and you feel that this manuscript is now acceptable for publication, you may indicate that here to bypass the “Comments to the Author” section, enter your conflict of interest statement in the “Confidential to Editor” section, and submit your "Accept" recommendation.

Reviewer #1: All comments have been addressed

2. Is the manuscript technically sound, and do the data support the conclusions?

Reviewer #1: Yes

3. Has the statistical analysis been performed appropriately and rigorously? 

Reviewer #1: N/A

4. Have the authors made all data underlying the findings in their manuscript fully available?

Reviewer #1: Yes

5. Is the manuscript presented in an intelligible fashion and written in standard English?

Reviewer #1: Yes

6. Review Comments to the Author

Reviewer #1: The authors appear to have appropriately addressed the reviewers' comments. I find the manuscript suitable for publication.

7. PLOS authors have the option to publish the peer review history of their article (what does this mean?). If published, this will include your full peer review and any attached files.

Reviewer #1: No

---

## [Editor Report · Acceptance letter]

16 Aug 2022

PONE-D-21-27892R1 

A Multidisciplinary Prematurity Research Cohort Study 

Dear Dr. England:

I'm pleased to inform you that your manuscript has been deemed suitable for publication in PLOS ONE. Congratulations! Your manuscript is now with our production department. 

Kind regards, 

on behalf of

Dr. Prem Singh Shekhawat 

Academic Editor

PLOS ONE